# Efficacy of the Adjunct Use of Povidone-Iodine or Sodium Hypochlorite with Non-Surgical Management of Periodontitis: A Systematic Review and Meta-Analysis

**DOI:** 10.3390/jcm11216593

**Published:** 2022-11-07

**Authors:** Marwan El Mobadder, Samir Nammour, Zuzanna Grzech-Leśniak, Kinga Grzech-Leśniak

**Affiliations:** 1Dental Surgery Department, Wroclaw Medical University, 50-425 Wroclaw, Poland; 2Department of Dental Sciences, Faculty of Medicine, University of Liege, 4000 Liege, Belgium

**Keywords:** periodontitis, periodontal therapy, non-surgical, povidone-iodine, sodium hypochlorite, NaOCl, PVP-I

## Abstract

This systematic review sought to assess the efficacy of combining either sodium hypochlorite or povidone-iodine as disinfection solutions with non-surgical treatment of periodontitis. An electronic search was conducted through PubMed, Scopus, Web of Science, CENTRAL, and Google Scholar from inception until 10 September 2022. Outcomes included clinical outcomes (probing pocket depth, plaque index, clinical attachment level, relative-horizontal attachment level, bleeding on probing, gingival recession, the position of gingival margin) and biochemical (BAPNA level) properties. A subgroup analysis was conducted according to the assessment timepoint. Ten studies reporting the use of povidone-iodine and five studies reporting the use of sodium hypochlorite were included in this review. Overall, in the meta-analysis of povidone-iodine, no significant changes were noted in any of the assessed outcomes; however, minor changes were noted in probing pocket depth and clinical attachment level at a specific timepoint. Regarding sodium hypochlorite, a significant reduction in all clinical outcomes, except for bleeding on probing, was noted. In conclusion, the use of povidone-iodine does not result in an improvement in clinical outcomes, whereas sodium hypochlorite has promising properties that result in significant improvement in probing pocket depth and clinical attachment level. However, more studies are needed to confirm these observations.

## 1. Introduction

Periodontitis is a multifactorial, biofilm-induced chronic inflammatory disease affecting and leading to the destruction of the tooth-supporting tissue and leading ultimately to tooth loss [1]. The progression of periodontitis depends largely on the availability and activity of specific pathogens found in the supra and subgingival areas [2,3,4]. For instance, these periodontopathogens were defined by Marsh et al. as the red complex and Hajishengallis et al. established the concept of keystone pathogens in periodontitis such as the *Porphyromonas gingivalis* [2,5,6]. The key goal of the non-surgical periodontal treatment remains to eliminate the supra and subgingival biofilm and to attenuate the inflammatory process [7]. Clinically, this manifests in a reduction of the pocket depth (PPD), an absence of bleeding on probing (BOP), and an increased attachment level [8,9]. Mechanical debridement consisting of manual and/or ultrasonic instrumentation remains the gold standard treatment [10]. However, due to some local and systematic limitations, such as the presence of deep periodontal pockets or deep furcation involvements, mechanical removal of subgingival calculus and biofilm is limited; thus, it does not always lead to the ultimate resolution of the inflammatory process [11,12,13]. This is why additional approaches that might increase disinfection are being suggested in the literature [14,15,16,17,18,19]. For example, disinfecting solutions [14], lasers [19], probiotics [17,18], and antimicrobial peptides [15,16] were suggested in numerous studies as adjuvants for SRP. As stated, the aim is to increase the overall bactericidal potential of the non-surgical treatment of Periodontitis [15,16,17,18,19,20].

Among these approaches, the use of sodium hypochlorite as a disinfection solution was evaluated in several studies [20,21,22,23,24,25]. Its broad antimicrobial activity and wild bactericidal action have been known and confirmed [20,21,22,23,24,25]. For instance, in root canal treatment, NaOCl is the standard irrigation solution of choice [26,27] due to its ideal antiseptic potential, broad spectrum, and rapid bactericidal effect. Moreover, in patients with periodontitis, a concentration of 0.1–0.5% of NaOCl has been used in numerous reports and found to be safe with a potential significant efficacy. These indications include essentially the non-surgical treatment of periodontitis in pocket depth greater than 6 mm and during periodontal surgery to disinfect the wound area with exposed alveolar bone [20,21,22,23,24,25].

On the other hand, Povidone-iodine (PVP-I) can be considered another promising disinfection solution. For instance, several studies have shown a beneficial impact of PVP-I when used conjunctly with subgingival debridement during the rinsing stage [28,29,30]; however, other studies have shown conflicting findings or no effect [31,32,33,34].

Hence, to the best of our knowledge, there is still no systematic review reporting the impact of sodium hypochlorite and povidone-iodine irrigation on the plaque index (PI), bleeding on probing (BOP), probing pocket depth (PPD), and clinical attachment level (CAL) in the non-surgical periodontal treatment (NSPT). Hence, the aim of this systematic review and meta-analysis is to review the effectiveness of the adjunctive subgingival application of sodium hypochlorite and povidone-iodine in the non-surgical treatment of periodontitis. PI, BOP, PPD, and CAL were the clinical parameters studied.

## 2. Materials and Methods

### 2.1. Study Design

This research was conducted in accordance with the Preferred Reporting Items for Systematic Reviews and Meta-Analyses (PRISMA) guidelines, where the pre-registration of a protocol is not mandated. The design of this research followed the PICOS framework as follows: population (patients with periodontitis of any severity and chronicity), intervention (the use of either NaOCl or PVP-I as an adjuvant to non-surgical periodontal therapy, comparison (non-surgical periodontal therapy alone), outcomes (clinical and biochemical parameters), and study design (randomized controlled trials “RCTs”).

### 2.2. Search Strategy

On 10 September 2022, PubMed, Scopus, Web of Science (WoS), Cochrane Central Register of Controlled Trials (CENTRAL), and Google Scholar were searched for RCTs that reported the efficacy of using either NaOCl or PVP-I as adjuvants to non-surgical periodontal therapy in patients with periodontitis. Noteworthy, based on recent recommendations [35], only the first 200 records of Google Scholar were searched, after which relevance significantly dropped. The following keywords were used to identify relevant articles: (periodontitis) AND (“sodium hypochlorite” OR “povidone-iodine”) AND “non-surgical”. Whenever possible, Medical Subject Headings (MeSH) terms were used to identify all potentially relevant articles. The search criteria were then adjusted based on the selected database. A full description of the search query used in each database is provided in Appendix A.

A manual search was also conducted following the screening of articles to identify any potentially missing relevant article through three approaches: (a) screening the reference list of included articles, (b) screening “similar articles” to included ones through the “similar articles” options on PubMed, and (c) manually searching for articles on Google with the use of following keywords: “periodontitis” + “non-surgical” + “sodium hypochlorite” or “povidone-iodine”.

### 2.3. Study Outcomes

The primary outcome included pro pocket depth (PPD), while secondary outcomes included plaque index (PI), clinical attachment level (CAL), relative horizontal attachment level (RHAL), bleeding on probing (BOP), gingival recession (GR), the position of gingival margin (PGM), and biochemical parameters such as N-benzoyl-L-arginine-p-nitroanilide level (BAPNA).

### 2.4. Eligibility Criteria

Studies were included when they met all of the following criteria: (1) randomized controlled trials, (2) including patients with periodontitis regardless of chronicity, severity, or classification used (Armitage’s classification or the 2017 classification by the European Federation of periodontology and the American Academy of Periodontology), (3) comparing either PVP-I + NSPT or NaOCl + NSPT to NSPT alone, and (4) reporting clinical outcomes (i.e., probing pocket depth, clinical attachment level, etc.). No limitation was set on language, publication date, country, or time of follow-up.

On the other hand, studies were excluded if they: (1) had no control group, (2) included patients with periodontal diseases other than periodontitis, (3) were not randomized, (4) included PVP-I or NaOCl in addition to other therapies, (5) reported irrelevant outcomes, (6) were non-original (i.e., reviews, editorials, letters, commentaries, etc.), (7) had no full text, or (8) were duplicate or contained overlapping datasets with other studies.

### 2.5. Study Selection

Following the retrieval of studies from the database search, citations were imported into EndNote for duplicate removal, after which citations were exported into an Excel Sheet for screening. First, the titles and abstracts of retrieved articles were screened against our prespecified eligibility criteria. Then, studies that were potentially relevant underwent full-text screenings. This process was carried out by two reviewers [M.E.M. and S.N.] who solved their differences through discussions. Meanwhile, [K.G.L.] was consulted when an agreement could not be reached. Noteworthy, if the full text of an article was not found, the authors of that article were contacted.

### 2.6. Data Extraction

A pilot extraction was carried out to design the data extraction sheet using Microsoft Excel. The data extraction sheet consisted of two parts. The first part included the baseline characteristics of included studies (first author’s name, year of publication, country, study design, and follow-up period) and patients (type and severity of periodontitis, intervention and control group, age, and gender]. The second part included the study outcomes [PPD, PI, CAL, RHAL, BAPNA, BOP, GR, and PGM). Noteworthy, BOP was extracted and analyzed as both a dichotomous and continuous outcome.

### 2.7. Risk of Bias Assessment

The risk of bias of included RCTs was assessed using the updated version of the Cochrane risk of bias (RoB-II) tool that was revised in 2019. This tool assesses the quality of RCTs at the level of five domains: randomization, deviation from intended protocols, missing outcomes data, measurement of the outcomes, and reporting of study findings. Each RCT was rated as having either low, some concerns, or a high risk of bias in each domain and collectively. This process was carried out by two reviewers who had prior training in using this sheet. Their results were compared to ensure an accurate assessment of the quality of included RCTs.

### 2.8. Data Synthesis

All statistical analyses were carried out through STATA Software (Version 17, College Station, TX, USA). The metan command was used to compare study outcomes between the intervention and comparison groups. The random-effects and fixed-effects models were selected according to the presence or absence of heterogeneity, respectively. Heterogeneity was present if the *I*^2^ statistic was above 50% and the *p*-value was <0.05. In outcomes where heterogeneity was encountered, the restricted maximum likelihood method (REML) was used; however, in the absence of heterogeneity, the inverse-variance (IV) or Mantel–Haenszel methods were used for continuous and dichotomous outcomes, respectively. A leave-one-out sensitivity analysis was carried out to determine if the reported effect estimate of each outcome was driven by a particular study. First, a meta-analysis was conducted based on the final assessment timepoints. Then, a subgroup analysis was conducted based on all of the reported assessment timepoints. The assessment of publication bias was not eligible due to the absence of the required number of studies (*n* = 10) in a single meta-analysis.

## 3. Results

### 3.1. Search Results

The database search and screening process is illustrated in Figure 1. A total of 903 articles were retrieved from the initial database search, out of which 121 were identified as duplicates through EndNote software and were then removed. The titles and abstracts of 782 were screened, yielding only 29 studies eligible for full-text screening. Upon retrieving the full texts, 16 articles were excluded for the following reasons: single-armed studies [*n* = 8], non-RCTs [*n* = 3], duplicated records [*n* = 2], interventions combined with other therapies such as tetracycline HCl or amino acid gel [*n* = 2] or reporting of irrelevant outcomes (microbiological data) [*n* = 1]. The manual search yielded two additional studies [36,37], while the updated database search resulted in no additional studies. Finally, a total of 15 RCTs were included in the qualitative and quantitative synthesis of this review [30,31,34,36,37,38,39,40,41,42,43,44,45,46,47].

### 3.2. Baseline Characteristics of Included Studies

The baseline characteristics of included studies, stratified by the type of intervention, are summarized in Table 1. A total of 15 RCTs were included (10 reporting the use of PVP-I and five reporting the use of NaOCl) in the final analysis. The overall sample size was 781 patients with periodontitis (520 in the PVP-I group and 261 in the NaOCl group). Data on the severity and chronicity of periodontitis can be found in Table 1.

In studies comparing adjuvant PVP-I to NSPT alone [30,31,34,38,40,42,43,44,46,47], 182 patients [43.40% males] received PVP-I in addition to the standard NSPT [reporting 3581 sites], while 215 patients [52.09% males] received the standard NSPT alone [reporting 3913 sites]. The follow-up duration ranged from 3 to 12 months.

In studies comparing adjuvant NaOCl to NSPT alone [36,37,39,41,45], 92 patients [47.82% males] received NaOCl in addition to the standard NSPT [reporting 1353 sites], while 95 patients [47.36% males] received the standard NSPT alone [reporting 1219 sites]. The follow-up duration ranged from 3 to 12 months.

### 3.3. Risk of Bias

The risk of bias assessment of included RCTs is provided in Table 2. Among studies comparing adjuvant PVP-I to NSPT alone, three RCTs [30,38,42] had a high risk of bias while the remaining had some concerns [31,34,40,43,44,46,47]. On the other hand, all of the RCTs comparing adjuvant NaOCl to NSPT alone had some concerns [36,37,39,41,45]. Noteworthy, the domains in which bias was encountered included randomization (either randomization was not carried out appropriately or there was no mention of the randomization method) and selective reporting (no prior registration of a trial protocol).

### 3.4. Meta-Analysis Outcomes [PVP-I + NSPT vs. NSPT Alone]

#### 3.4.1. Probing Depth of Pockets

Nine studies were included in the meta-analysis of the PPD. Overall, no significant difference in the PPD was noted between PVP-I + NSPT as compared to NSPT alone [MD = −0.31; 95%CI: −0.31: 0.06; *I*^2^ = 98.07%] [Figure 2]. No significant change in the reported effect estimate was noted following the exclusion of one study at a time through a sensitivity analysis.

Additionally, no significant difference between both groups was noted at 1, 3, or 6 months [Appendix A]. However, during the 12th month, a significant reduction in PPD was noted in the PVP-I + NSPT group as compared to the NSPT alone group [MD = −0.16; 95%CI: −0.23: −0.09; *I*^2^ = 58.75%].

#### 3.4.2. Plaque Index

Two studies reporting the PI were included in this meta-analysis. Overall, no significant change in the PI was noted between both groups [MD= −3.41; 95%CI: −9.06: 2.25; *I*^2^ = 0%] [Figure 3]. No significant change in the reported effect estimate was noted following the exclusion of one study at a time through a sensitivity analysis.

Upon doing a subgroup analysis based on the assessment timepoint, no significant differences were found between both groups at 1, 3, and 6 months [Appendix A].

#### 3.4.3. Clinical Attachment Level

Eight studies reporting the CAL were included in this meta-analysis. Overall, no significant difference between both treatment groups was noted [MD = −0.11; 95%CI: −0.28: 0.05; *I*^2^ = 87.18%] [Figure 4]. No significant change in the reported effect estimate was noted following the exclusion of one study at a time through a sensitivity analysis.

Upon doing a subgroup analysis based on the assessment timepoint, no significant differences were found between PVP-I + NSPT and NSPT alone at 1 and 3 months [Appendix A]. However, a significant reduction in the CAL was noted in the PVP-I + NSPT group at 6 months [MD = −0.20; 95%CI: −0.30: −0.10; *I*^2^ = 0%].

#### 3.4.4. Relative Horizontal Attachment Level

Two studies reporting the RHAL were included in this meta-analysis, revealing no significant change between both groups [MD = 0.43; 95%CI: −0.26: 1.12; *I*^2^ = 0%] [Figure 5]. No significant change in the reported effect estimate was noted following the exclusion of one study at a time through a sensitivity analysis.

Upon doing a subgroup analysis based on the assessment timepoint, no significant differences were found between both groups at 1, 3, and 6 months [Appendix A].

#### 3.4.5. Bleeding on Probing

Three studies reported BOP as a dichotomous outcome. The meta-analysis of these studies revealed no significant change between both groups [LogOR = −0.37; 95%CI: −0.90: 0.16; *I*^2^ = 0%] [Figure 6].

This is also consistent with the findings of the meta-analysis of BOP as a continuous outcome [MD = 0.30; 95%CI: −0.46: 1.07; *I*^2^ = 0%] [Figure 7]. No significant change in the reported effect estimate was noted following the exclusion of one study at a time through a sensitivity analysis.

Upon doing a subgroup analysis based on assessment timepoint, no significant difference in the risk of developing BOP was found between both groups at 1, 3, and 6 months [Appendix A], which was confirmed in the analysis of BOP as a continuous outcome [Appendix A].

#### 3.4.6. Gingival Recession

Three studies reporting GR were included in this meta-analysis revealing no significant change between both groups [MD = 0.14; 95%CI: −0.10: 0.38; *I*^2^ = 0%] [Figure 8]. No significant change in the reported effect estimate was noted following the exclusion of one study at a time through a sensitivity analysis.

Upon doing a subgroup analysis based on the assessment timepoint, no significant differences were found between both groups at 1 and 3 months [Appendix A].

#### 3.4.7. Position of Gingival Margin

Two studies reporting the PGM were included in this meta-analysis, showing no significant difference between both treatment arms [MD = −0.23; 95%CI: −0.69: 0.24; *I*^2^ = 0%] [Figure 9]. No significant change in the reported effect estimate was noted following the exclusion of one study at a time through a sensitivity analysis.

Upon doing a subgroup analysis based on the assessment timepoint, no significant differences were found between both groups at 1, 3, and 6 months [Appendix A].

#### 3.4.8. Biochemical Parameter: BAPNA

Two studies reporting BAPNA were included in this meta-analysis. Overall, no significant change in this outcome was noted between both groups [MD = −8.69; 95%CI: −29.50: 12.12; *I*^2^ = 67.77%] [Figure 10]. Notably, the confidence interval is too wide to drive any conclusions. No significant change in the reported effect estimate was noted following the exclusion of one study at a time through a sensitivity analysis.

Upon doing a subgroup analysis based on the assessment timepoint, no significant differences were found between both groups at 1, 3, and 6 months [Appendix A].

### 3.5. Meta-Analysis Outcomes [NaOCl + NSPT vs. NSPT Alone]

#### 3.5.1. Probing Depth of Pockets

Five studies reporting the PPD were meta-analyzed revealing a barely significant change between NaOCl + NSPT and NSPT alone [MD = −0.22; 95%CI: −0.43: 0.00; *I*^2^ = 85.60%] [Figure 11]. The leave-one-out sensitivity analysis revealed no significant change in the reported effect estimate upon removing one study at a time.

No significant difference between both groups was noted at 1, 3, or 12 months [Appendix A]. However, at 6 months, a significant reduction in the PPD was noted in the NaOCl + NSPT group as compared to the NSPT alone group [MD = −0.29; 95%CI: −0.57: −0.01; *I*^2^ = 92.43%].

#### 3.5.2. Clinical Attachment Level

Four studies were meta-analyzed, revealing a significant reduction of the CAL in the NaOCl + NSPT group as compared to the NSPT alone group [MD = −0.44; 95%CI: −0.83: −0.05; *I*^2^ = 89.38%] [Figure 12]. The leave-one-out sensitivity analysis revealed no significant change in the reported effect estimate upon removing one study at a time.

Upon doing a subgroup analysis based on the assessment timepoint, no significant differences were found between both groups at 1, 2, 6, 9, and 12 months [Appendix A]. However, a significant reduction in the CAL was noted in the NaOCl + NSPT group at 3 months [MD = −0.32; 95%CI: −0.50: −0.14; *I*^2^ = 31.69%].

#### 3.5.3. Bleeding on Probing

Two studies were meta-analyzed, revealing no significant change in BOP between both groups [LogOR = −0.32; 95% CI: −1.39: 0.74; *I*^2^ = 83.76%] [Figure 13]. The leave-one-out sensitivity analysis revealed no significant change in the reported effect estimate upon removing one study at a time.

Upon doing a subgroup analysis based on the assessment timepoint, no significant difference in the risk of developing BOP was found between both groups at 3, 6, 9, or 12 months [Appendix A]. Notably, at 3, 6, and 9 months, only one study was included; therefore, no conclusions can be drawn from the analysis of these timepoints.

#### 3.5.4. Gingival Recession

Two studies reporting GR were meta-analyzed, reporting a significant reduction in GR in the NaOCl + NSPT group as compared to the NSPT alone group [MD = −0.24; 95%CI: −0.42: −0.07; *I*^2^ = 0%] [Figure 14]. However, the sensitivity analysis revealed no significant change in the reported outcome following the exclusion of the study of Iorio-Siciliano et al. [41] [Appendix A].

Notably, since the meta-analysis was based on only two studies, no applicable conclusions can be drawn from this finding. A subgroup analysis based on the assessment timepoint was not feasible due to the lack of relevant data.

#### 3.5.5. Other Parameters

No studies compared NaOCl + NSPT to NSPT alone in terms of RHAL, PGM, and BAPNA. Of note, only one study reported the PI and BOP [39]; therefore, a meta-analysis could not be conducted.

## 4. Discussion

The process of reducing or eliminating the biofilm in infected periodontal tissues has been extensively studied, reporting the importance of complex inner anatomy, the dynamic relation of the microorganism, and factors related to host response [48]. Mechanical debridement, through NSPT alone, can in some cases result in limited effectiveness as a standalone treatment to completely treat periodontitis and for instance, reduce pockets that are greater than 6 mm [49]. Therefore, the adjunct use of chemical products has been proposed, assuming an added benefit in clinical and microbiological outcomes, especially if these proposed disinfection solutions were effective on the periodonto-pathogens.

This present systematic review and meta-analysis assessed the clinical efficacy of using sodium hypochlorite or povidone-iodine as adjunctive to non-surgical treatment of periodontitis which might prove to be beneficial to use in periodontal pockets with a depth greater than 6 mm after the non-surgical treatment. In fact, the introduction of additional approaches might lead to a better outcome of PPD and CAL with non-surgical treatment and thus avoiding surgical intervention [7].

Sodium hypochlorite is used worldwide as a root canal irrigating agent in endodontic treatment due essentially to its effectiveness for its organic dissolution and also to its antimicrobial activity [48]. Hypochlorous acid, a substance present in sodium hypochlorite solution, when in contact with organic tissue acts as a solvent and releases chlorine that, combined with the protein amino group, forms chloramines [48]. Hypochlorous acid (HOCl-) and hypochlorite ions (OCl-) lead to amino acid degradation and hydrolysis [48]. This chloramination reaction between chlorine and the amino group (NH) forms chloramines that interfere with cell metabolism [48,50]. Chlorine (strong oxidant) presents a strong antimicrobial action due to its potential to inhibit bacterial enzymes leading thus to irreversible oxidation of SH groups (sulphydryl group) of essential bacterial enzymes [48,50]. Since periodontitis is a biofilm-induced chronic infection, the use of sodium hypochlorite in order to increase disinfection is explainable.

Our meta-analysis of five RCTs revealed a statistically significant reduction in PPD in favor of sodium hypochlorite; however, the clinical significance of this finding should be carefully interpreted since the confidence interval is almost at the null value. This difference was maintained at 1, 2, 6, and 9 months, while other assessment timepoints (3 and 12 months) revealed no significant change. Notably, the findings based on data reported at 1, 2, and 9 months are based only on one study, so more studies are still warranted to confirm this observation. Sodium hypochlorite also showed a significant reduction in CAL as compared to the standalone NSPT which was observed only after 3 months of therapy. Due to the lack of a sufficient number of studies, no conclusions can be drawn on whether or not the added benefit of sodium hypochlorite can be maintained in the long term. Although it revealed no significant change in BOP, sodium hypochlorite resulted in a significant reduction in GR.

Povidone-iodine, a known antiseptic solution with broad bactericidal properties, was reported to be used in conjunction with NSPT in periodontitis patients. Although some studies have shown a beneficial effect of using PVP-I as an add-on to NSPT, other studies reported contradictory findings. In our systematic review of PVP-I, a total of 10 studies were included and analyzed, revealing a non-significant change in almost all of the assessed clinical and biochemical parameters. A previous systematic review highlighted a significant, yet minor, reduction in PPD with the adjuvant use of PVP-I to NSPT in the treatment of chronic periodontitis; however, in the subgroup analysis based on follow-up, no significant changes were noted [51]. Controversially, our study highlighted no significant change in PPD with the use of PVP-I compared to the use of NSPT as a standalone. However, it should be noted that we encountered a significant considerable heterogeneity (98.07%, *p* = 0.00) which could affect the reliability of this observation. Upon doing a subgroup analysis, we noted that the assessment timepoint was a significant contributor to the observed heterogeneity, and then at 12 months a statistically significant reduction in PPD was observed (heterogeneity = 58.75%, *p* = 0.12). However, it should be noted that this change in the effect size is low based on Cohen’s classification (MD = −0.16; 95%CI: −0.23: −0.09). This could indicate that PVP-I might not be effective in the short term, but it can result in a significant change in the long term. That being said, future, well-designed RCTs are still needed to confirm this finding.

In our meta-analysis, a very similar observation to PPD was noted regarding CAL, where no significant change was noted in the overall meta-analysis; however, a significant reduction in CAL was documented after 6 months of therapy. Unfortunately, PVP-I did not show any resolution of inflammation or plaque control since its use did not result in any significant differences in either plaque index or bleeding on probing as compared to NSPT alone. In addition, no significant change was noted in the remaining clinical (gingival recession or position of gingival margin) or biochemical (BAPNA levels) outcomes.

At present, 0.12% chlorhexidine (CHX) is the most acceptable chemical agent to assess mechanical debridement [7,52]. The recently published clinical guideline of EFP and AAP for the treatment of periodontitis suggest only the use of CHX as an irrigation solution [7]. A very important difference between CHX and NaOCl or PVP-I is that the application of CHX can effectively prevent biofilm formation after mechanical debridement. Our findings highlight that PVP-I cannot replace CHX as a standard irrigation solution; however, the use of NaOCl might have an added benefit regarding a number of clinical outcomes (PPD, CAL, and GR). That being said, we are unable to confirm whether these effects can be reached at particular concentrations or follow-up times. Therefore, we invite higher-quality RCTs with a longer period of follow-up and bigger sample size to study the actual effectiveness of NaOCl or PVP-I as an adjunct to the non-surgical treatment of periodontitis.

### Study Limitations and Future Directions

Although our study provides valuable insights into the concurrent management of periodontitis either with NaOCl or PVP-I in addition to NSPT, our study has several limitations. First, the current meta-analysis does not compare the efficacy between NaOCl or PVP-I due to the lack of direct- or indirect-comparison studies which made it impossible to conduct a meta-analysis or a network meta-analysis, respectively. Second, significant heterogeneity was encountered in a number of analyses which sometimes reflected a change based in the timepoint; however, the different application modalities and concentrations of either NaOCl or PVP-I could attribute to the observed heterogeneity. Moreover, included studies recruited patients with variable classifications of periodontitis, and this could have played a role in the encountered heterogeneity. Third, we did not assess the impact of either intervention on the microbiological outcomes due to the lack of a sufficient number of studies reporting relevant data. Fourth, the majority of included RCTs had some concerns or a high risk of bias. Fifth, the confidence interval of some outcomes (i.e., BAPNA) was quite wide, reflecting the imprecision in the reported effect estimate. Therefore, our findings should be carefully interpreted. Additionally, more studies are warranted to determine if the effect of either intervention (NaOCl or PVP-I) would differ based on the depth of examined roots.

## 5. Conclusions

Based on the evidence of this systematic review, the use of povidone-iodine as an adjunctive disinfectant solution for the non-surgical treatment of periodontitis does not present an additional benefit in terms of clinical outcomes; however, using sodium hypochlorite resulted in an added benefit compared to standalone non-surgical periodontal therapy. However, more studies of better quality are still needed.

## Figures and Tables

**Figure 1 jcm-11-06593-f001:**
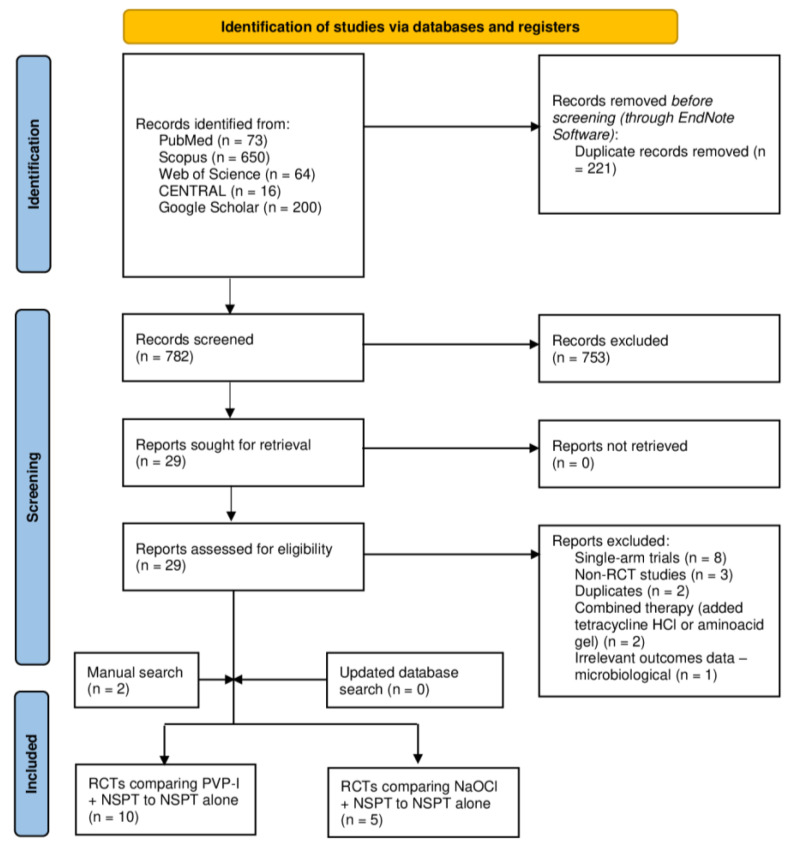
A PRISMA flow diagram of the database search and screening processes. RCT: randomized clinical trial. PVP-1: Povidone-Iodine. NSPT: non-surgical periodontal treatment. NaOCL: sodium hypochlorite.

**Figure 2 jcm-11-06593-f002:**
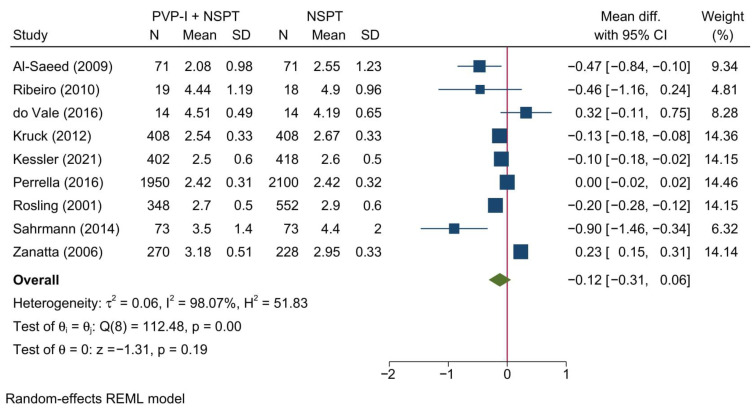
Forest plot showing the comparison between PVP-I + NSPT and NSPT alone regarding probing pocket depth [30,34,38,40,42,43,44,46,47].

**Figure 3 jcm-11-06593-f003:**
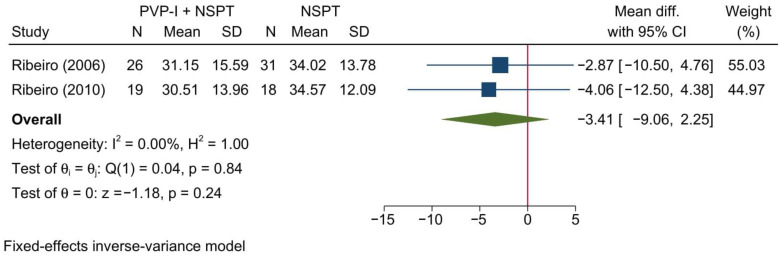
Forest plot showing the comparison between PVP-I + NSPT and NSPT alone regarding plaque index [31,46].

**Figure 4 jcm-11-06593-f004:**
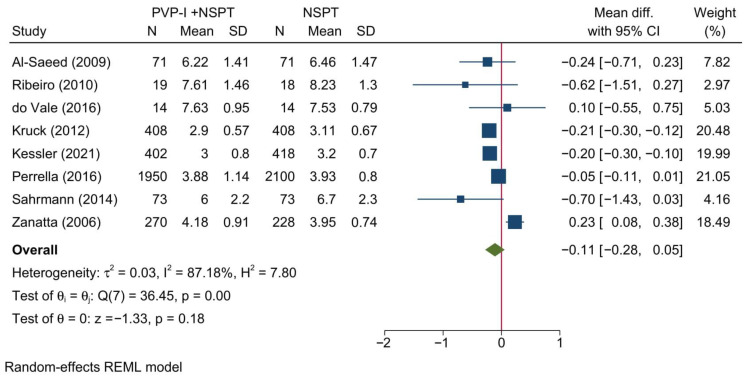
Forest plot showing the comparison between PVP-I + NSPT and NSPT alone regarding clinical attachment level [34,38,40,42,43,44,46,47].

**Figure 5 jcm-11-06593-f005:**
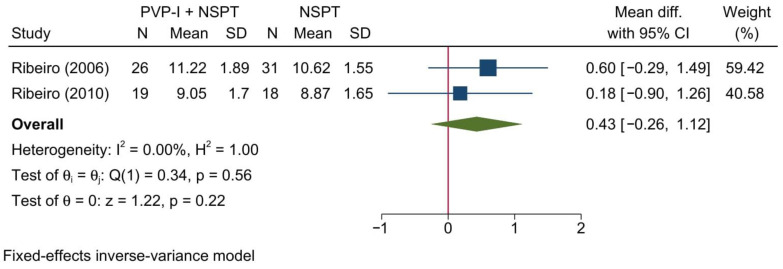
Forest plot showing the comparison between PVP-I + NSPT and NSPT alone regarding relative horizontal attachment level [31,46].

**Figure 6 jcm-11-06593-f006:**
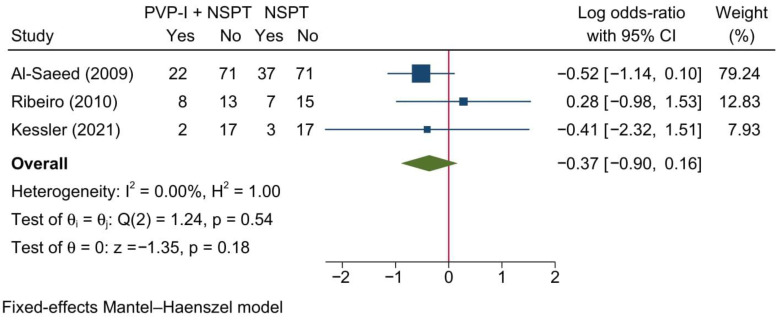
Forest plot showing the comparison between PVP-I + NSPT and NSPT alone regarding bleeding on probing (dichotomous variable) [38,42,46].

**Figure 7 jcm-11-06593-f007:**
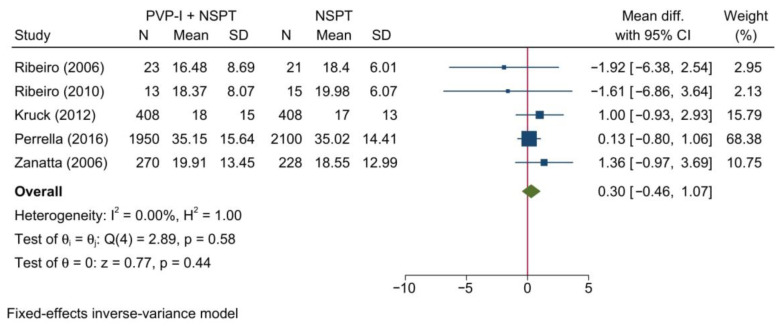
Forest plot showing the comparison between PVP-I + NSPT and NSPT alone regarding bleeding on probing (continuous variable) [31,34,43,44,46].

**Figure 8 jcm-11-06593-f008:**
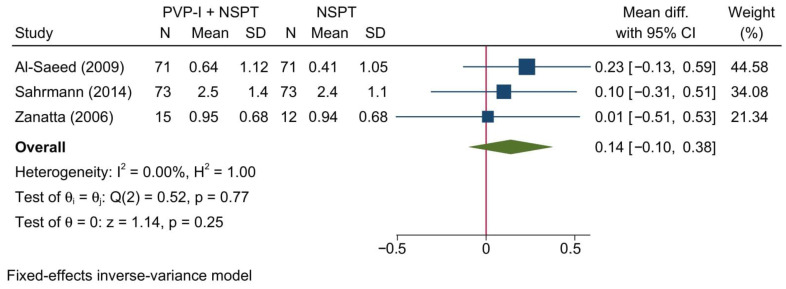
Forest plot showing the comparison between PVP-I + NSPT and NSPT alone regarding gingival recession [34,38,47].

**Figure 9 jcm-11-06593-f009:**
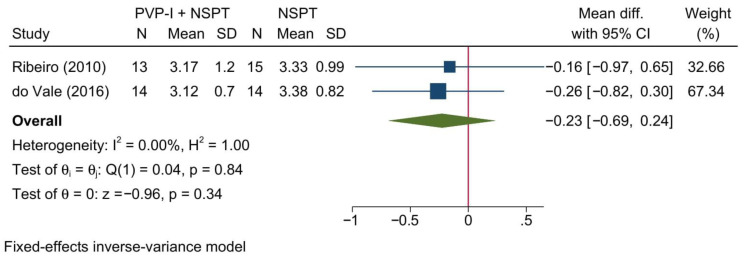
Forest plot showing the comparison between PVP-I + NSPT and NSPT alone regarding the position of gingival margin [40,46].

**Figure 10 jcm-11-06593-f010:**
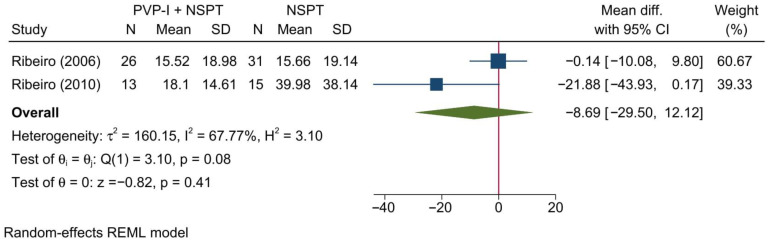
Forest plot showing the comparison between PVP-I + NSPT and NSPT alone regarding BAPNA [31,46].

**Figure 11 jcm-11-06593-f011:**
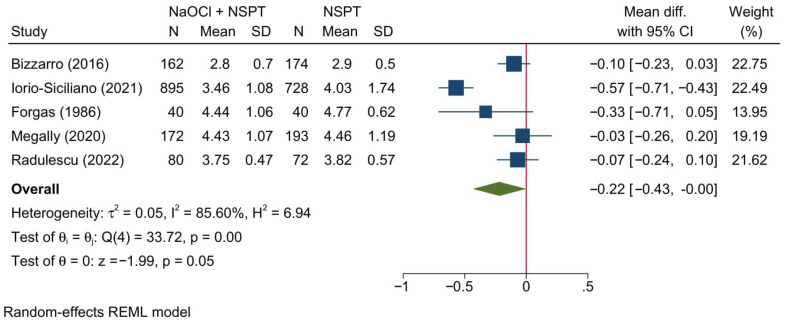
Forest plot showing the comparison between NaOCl + NSPT and NSPT alone regarding probing pocket depth [36,37,39,41,45].

**Figure 12 jcm-11-06593-f012:**
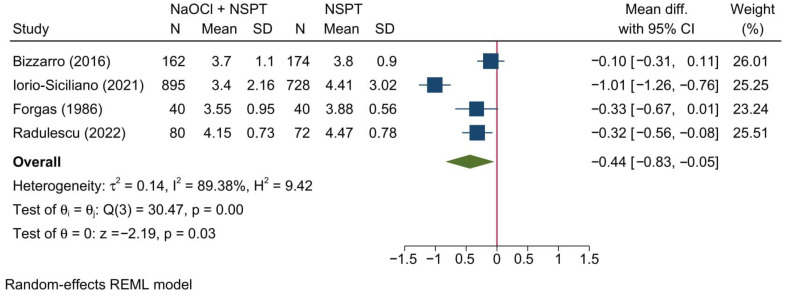
Forest plot showing the comparison between NaOCl + NSPT and NSPT alone regarding clinical attachment level [36,39,41,45].

**Figure 13 jcm-11-06593-f013:**
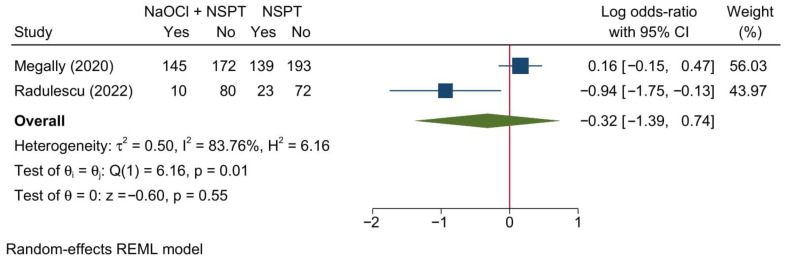
Forest plot showing the comparison between NaOCl + NSPT and NSPT alone regarding bleeding on proving (dichotomous) [37,45].

**Figure 14 jcm-11-06593-f014:**
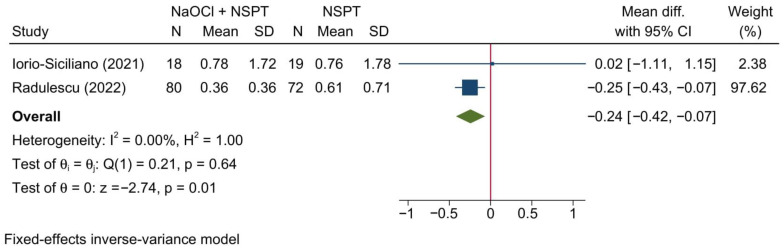
Forest plot showing the comparison between NaOCl + NSPT and NSPT alone regarding gingival recession [41,45].

**Table 1 jcm-11-06593-t001:** Baseline characteristics of included studies reporting the use of PVP-I or NaOCl as an adjuvant to non-surgical debridement in periodontitis (*n* = 14).

Author (YOP)	Country	Design	Sample (*n*)	Periodontitis		Intervention			Control			Gender [Male]		Age				Follow-up (months)
				Severity	Chronicity	*n*	Sites	Description	*n*	Sites	Description	Intervention	Control	Intervention		Control		
												*n* (%)	*n* (%)	Mean	SD	Mean	SD	
**PVP-I + NSPT vs. NSPT Alone**																		
Al-Saeed (2009) [38]	Saudi Arabia	Single-blinded RCT	26	Slight to Moderate [*n* = 13]	Chronic [*n* = 13]	-	71	PVP-I + ultrasonic irrigation + root planning	-	71	Ultrasonic irrigation + root planning alone	-	-	-	-	-	-	3
Ribeiro (2006) [31]	Brazil	Single-blinded, parallel-arm RCT	44	-	-	23	26	PVP-I + subgingival instrumentation	21	31	subgingival instrumentation	11 (47.83%)	9 (42.86%)	42.96	-	42.52	-	6
Ribeiro (2010) [46]	Brazil	Parallel-arm RCT	28	-	-	13	19	PVP-I + subgingival instrumentation	15	18	subgingival instrumentation	6 (46.15%)	7 (47.37%)	43.74	-	42.96	-	6
do Vale (2016) [40]	Brazil	Parallel-arm RCT	34	Aggressive [*n* = 34]	Chronic [*n* = 34]	14	14	FMUD + PVPI	14	14	FMUD + SS	2 (14.29%)	5 (35.71%)	28.54	4.14	28.57	4.59	6
Kessler (2021) [42]	Belgium	Single-blinded RCT	34	-	Chronic [*n* = 17]	17	402	SRP + PVP-I	17	418	SRP + SS	9 (52.94%)	9 (52.94%)	51.8	-	51.8	-	12
Kruck (2012) [43]	Germany	RCT	51	Moderate [*n* = 51]	Chronic [*n* = 51]	17	408	SRP + PVP-I	17	408	SRP + NaCl	8 (47.06%)	14 (41.18%)	50.13	9.74	51.82	10.61	6
Perrella (2016) [44]	Brazil	RCT	29	-	Chronic [*n* = 29]	14	1950	SRP + PVP-I	15	2100	SRP + irrigation with saline solution	5 (35.71%)	7 (46.67%)	43.93	3.13	44.87	4.41	3
Rosling (2001) [30]	USA	RCT	223	Advanced destructive [*n* = 223]	Chronic [*n* = 223]	58	348	Non-surgical ultrasonic instrumentation + PVP-I	92	552	Non-surgical ultrasonic instrumentation	29 (50.6%)	52 (56.60%)	44.2	8.8	44.5	8.6	12
Sahrmann (2014) [47]	Switzerland	RCT	11	Severe [*n* = 11]	Chronic [*n* = 11]	11	73	SRP + PVP-I	11	73	SRP + water	9 (81.81%)	9 (81.81%)	48.9	-	48.9	-	3
Zanatta (2006) [34]	Brazil	Single-blinded, parallel-arm RCT	40	Moderate [*n* = 40]	Chronic [*n* = 40]	15	270	Non-surgical ultrasonic instrumentation + PVP-I	13	228	Root planing + NaCl irrigation	-	-	42	-	40	-	3
**NaOCl + NSPT vs. NSPT Alone**																		
Bizzarro (2016) [39]	Netherlands,	RCT	110	-	-	27	162	Basic periodontal therapy + NaOCl	29	174	Basic periodontal therapy + Saline	15 (55.00%)	11 (44.00%)	47.7	11.2	46.9	8.5	12
Iorio-Siciliano (2021) [41]	Italy	RCT	37	Severe [*n* = 37]	Chronic [*n* = 37]	18	895	MINST and NaOCl gel application	19	728	MINST alone	6 (33.33%)	10 (52.63%)	53.3	9.8	48.5	6.5	6
Forgas (1986) [36]	USA	RCT	20	Moderate [*n* = 20]	-	10	40	SRP + NaOCl	10	40		6 (60.00%)	6 (60.00%)	Both groups [mean = 38.8]				3
Megally (2020) [37]	Switzerland	RCT	32	-	-	16	172	NaOCl gel + ultrasonic debridement	16	193	ultrasonic debridement only	7 (43.75%)	6 (37.50%)	61.7	9.8	62.1	8.8	12
Radulescu (2022) [45]	Romania	Triple-blinded RCT	62	Stage III-IV [*n* = 62]	-	21	84	NaOCl gel + mechanical re-instrumentation	21	84	Placebo gel + mechanical re-instrumentation	10 (50.00%)	12 (66.66%)	44.6	9.86	50.61	9.31	12

*n*: Number of patients; YOP: Year of publication; PVP-I: Povidone-iodine; NaOCl: Sodium hypochlorite; SRP: Scaling and root planing; SS: Single session; FMUD: Full-mouth ultrasonic debridement; MINST: Minimally-invasive non-surgical periodontal therapy; SD: Standard deviation; RCT: Randomized controlled trial.

**Table 2 jcm-11-06593-t002:** The risk of bias of included RCTs based on the revised version of Cochrane’s risk of bias tool.

Author (YOP)	Randomization	Deviations from Intended Interventions	Missing Outcomes Data	Outcome Measurement	Selective Reporting	Overall
**PVP-I + NSPT vs. NSPT Alone**						
Al-Saeed (2009) [38]	Some concerns	Low	High	Low	Some concerns	High
Del Peloso Ribeiro (2006) [31]	Some concerns	Low	Low	Low	Some concerns	Some concerns
Del Peloso Ribeiro (2010) [36]	Low	Low	Some concerns	Low	Some concerns	Some concerns
do Vale (2016) [40]	Some concerns	Low	Low	Low	Some concerns	Some concerns
Kruck (2012) [43]	Some concerns	Low	Low	Low	Some concerns	Some concerns
Kessler (2021) [42]	Some concerns	High	Some concerns	Low	Some concerns	High
Perrella (2016) [44]	Some concerns	Low	Low	Low	Some concerns	Some concerns
Rosling (2001) [30]	Some concerns	High	Low	Low	Some concerns	High
Sahrmann (2014) [47]	Some concerns	Low	Low	Low	Some concerns	Some concerns
Zanatta (2006) [34]	Some concerns	Low	Low	Low	Some concerns	Some concerns
**NaOCl + NSPT vs. NSPT Alone**						
Bizzarro (2016) [39]	Low	Low	Low	Low	Some concerns	Some concerns
Iorio-Siciliano (2021) [41]	Low	Low	Low	Low	Low	Some concerns
Forgas (1986) [36]	Some concerns	Low	Low	Low	Some concerns	Some concerns
Megally (2020) [37]	Some concerns	Low	Low	Low	Some concerns	Some concerns
Radulescu (2022) [45]	Low	Low	Low	Low	Some concerns	Some concerns

RoB: Risk of Bias; YOP: Year of Publication; NSPT: Non-Surgical Periodontal Therapy; PVP-I: Povidone-Iodine; NaOCl: Sodium Hypochlorite.

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
