# Peer review of "Efficacy of the Adjunct Use of Povidone-Iodine or Sodium Hypochlorite with Non-Surgical Management of Periodontitis: A Systematic Review and Meta-Analysis"

_jcm, 2022, doi:10.3390/jcm11216593_

Round 1
Reviewer 1 Report
In this review, the authors aimed to assess the efficacy of the adjunct use of povidone-iodine or sodium hypochlorite with non-surgical management of periodontitis. Using different databases (PubMed, Scopus, Web of Science, CENTRAL, and Google Scholar), the metanalysis showed that povidone-iodine no changes the outcome of the disease. By contrast, sodium hypochlorite reduced all clinic outcomes, except for bleeding on probing. Although the manuscript does not describe any novel results for clinicians, it can serve as a summary of clinical data.
Author Response
Dear reviewer. We would like to thank you for your time and for considering our systematic review and meta-analysis worth publishing. We agree that this study can serve as a summary of the clinical data.
Reviewer 2 Report
Dear Editor,
Regarding the submitted manuscript “ Efficacy of the Adjunct Use of Povidone-Iodine or Sodium Hypochlorite with Non-Surgical Management of Periodontitis: A Systematic Review and Meta-Analysis” this review will be divided in overall and detailed appreciation.
The presented study is intended to be a systematic written according to the Prisma Statement. In order to perform a structured analysis of the manuscript the ROBIS checklist for evaluating SR will be used.
ROBIS Detailed appreciation
Phase 1: Assessing relevance
The authors presented the question according to PICOT criteria.
Phase 2: Identifying concerns with the review process
DOMAIN 1: STUDY ELIGIBILITY CRITERIA
Describe the study eligibility criteria, any restrictions on eligibility and whether there was evidence that objectives and eligibility criteria were pre-specified.
While addressing the stated points in the ROBIS tools the concerns regarding specification of study eligibility criteria of this referee are:
Although not mandatory, the study protocol should be registered (ex: Prospero) since enables the readers to check for additional details and also to confirm that the study design was made à priori from the obtained results.
The eligibility criteria are not clear.
Months/Year in the time limits instead of just years (point 6 prisma)
The search should be performed without language restriction (point 6 prisma) – the authors do not mention languages
Time of follow up for eligibility
Grey literature search needs to be performed and the authors of the included studies contacted (point 7 prisma) – The authors mentioned that they search the first 100 references according to the study with reference 35, but the article mentions that the 200 first articles should be searched. Please correct it
DOMAIN 2: IDENTIFICATION AND SELECTION OF STUDIES
Regarding the study itself it is advisable redoing the search adding the following criteria to prevent the selection bias:
- Contacting all the authors of the included studies to request for additional published or unpublished data
- Supplementary Table 1 needs to be provided
DOMAIN 3: DATA COLLECTION AND STUDY APPRAISAL
- Study appraisal calibration and evaluation need to be performed (kappa between reviewers in the critical appraisal)
DOMAIN 4: SYNTHESIS AND FINDINGS
The synthesis and findings is well performed
Author Response
Phase 1: Assessing relevance
The authors presented the question according to PICOT criteria.
Phase 2: Identifying concerns with the review process
DOMAIN 1: STUDY ELIGIBILITY CRITERIA
Describe the study eligibility criteria, any restrictions on eligibility and whether there was evidence that objectives and eligibility criteria were pre-specified.
While addressing the stated points in the ROBIS tools the concerns regarding specification of study eligibility criteria of this referee are:
- Although not mandatory, the study protocol should be registered (ex: Prospero) since enables the readers to check for additional details and also to confirm that the study design was made à priori from the obtained results.
Response: Thank you for your comment. However, as you mentioned, the registration of the protocol is not mandatory, and at this point, registering it would make no difference and it would take between 2-3 weeks until it gets approved.
- The eligibility criteria are not clear. Months/Year in the time limits instead of just years (point 6 prisma).
Response: Thank you for the comment. The timing in which the database was performed has been reported in subsection “search strategy”. Studies that were published before Sept 10, 2022 were included, as long as they satisfied our eligibility criteria.
- The search should be performed without language restriction (point 6 prisma) – the authors do not mention languages.
Response: Thank you for highlighting this point. However, in the original manuscript we reported “No limitation was set on language, publication date, or country”. We included all studies regardless of their language.
- Time of follow up for eligibility.
Response: Thank you for your comment. We added a statement about this in the revised manuscript as follows: “No limitation was set on language, publication date, country, or time of follow-up”. Since we were interested in determining whether the treatment effect would be modified according to the assessment timepoint, we included all RCTs regardless of their follow up time.
- Grey literature search needs to be performed and the authors of the included studies contacted (point 7 prisma) – The authors mentioned that they search the first 100 references according to the study with reference 35, but the article mentions that the 200 first articles should be searched. Please correct it.
Response: Thank you for highlighting this mistake. We corrected this in the revised version of the manuscript (200 references instead of 100). As for the grey literature, we did search Google scholar for this purpose. To clarify, we could not access the full text of the study of Radulescu et al. and we contacted the authors through ResearchGate and they provided us with their manuscript, which we further included in our review. We added a sentence in the revised manuscript to clarify this point as follows: “Noteworthy, if the full text of an article was not found, the authors of that article were contacted.”
DOMAIN 2: IDENTIFICATION AND SELECTION OF STUDIES
- Regarding the study itself it is advisable redoing the search adding the following criteria to prevent the selection bias: Contacting all the authors of the included studies to request for additional published or unpublished data
Response: Thank you for your comment. We carried out several methods to minimize selection bias: (1) an exhaustive database search was performed through four databases (PubMed, Scopus, Web of Science, and CENTRAL), (2) we included Grey Literature (Google Scholar) as well to avoid publication bias, (3) the screening was done by 2 reviewers and the third reviewer revised all of their work to ensure the inclusion of all relevant articles, (4) in instances when the full text of a manuscript couldn’t be found, the authors of that paper were contacted, and (5) assessment of publication bias was not feasible because of the low number of included RCTs (<10 in each analysis).
- Supplementary Table 1 needs to be provided.
Response: Thank you for your comment. We must have forgotten to upload it with the initial submission. However, we uploaded it with the revised version of the manuscript.
DOMAIN 3: DATA COLLECTION AND STUDY APPRAISAL
- Study appraisal calibration and evaluation need to be performed (kappa between reviewers in the critical appraisal)
Response: Thank you for your comment. Before the risk of bias assessment step, both reviewers were given training in how to assess the risk of bias of RCTs following Cochrane’s guide with graphical representations to highlight which decisions should be made regarding each domain in the risk of bias sheet. The reporting of both reviewers was compared and it was identical; we did not find any differences between them. We added a sentence in the manuscript to support that as follows: “This process was carried out by two reviewers who had prior training in using this sheet. Their results were compared to ensure accurate assessment of the quality of included RCTs.”
DOMAIN 4: SYNTHESIS AND FINDINGS
- The synthesis and findings is well performed.
Response: Thank you for your tremendous help in improving the quality of our manuscript.
Reviewer 3 Report
Excellent sistematic review about the efficacy of the Adjunct Use of Povidone-Iodine or Sodium Hypochlorite with Non-Surgical Management of Periodontitis. This systematic review sought to assess the efficacy of combining either sodium hypo- 11 chlorite or povidone-iodine as disinfection solutions with non-surgical treatment of periodontitis.
Authors have carried out a coorect elegibility criteria, study selection and data extraction.
The article is very well organized and written.
Author Response
Dear reviewer, we would like to thank you for your positive comments and feedback and for considering our paper worth publishing.